# Cytokine-Laden Extracellular Vesicles Predict Patient Prognosis after Cerebrovascular Accident

**DOI:** 10.3390/ijms22157847

**Published:** 2021-07-22

**Authors:** Anthony Fringuello, Philip D. Tatman, Tadeusz Wroblewski, John A. Thompson, Xiaoli Yu, Kevin O. Lillehei, Robert G. Kowalski, Michael W. Graner

**Affiliations:** 1Department of Neurosurgery, Anschutz Medical Campus, University of Colorado, Aurora, CO 80045, USA; anthony.fringuello@gmail.com (A.F.); philip.tatman@cuanschutz.edu (P.D.T.); tadeusz.wroblewski@cuanschutz.edu (T.W.); john.a.thompson@cuanschutz.edu (J.A.T.); xiaoli.yu@cuanschutz.edu (X.Y.); KEVIN.LILLEHEI@CUANSCHUTZ.EDU (K.O.L.); robert.2.kowalski@cuanschutz.edu (R.G.K.); 2Department of Neurology, Anschutz Medical Campus, University of Colorado, Aurora, CO 80045, USA; 3Medical Scientist Training Program, Anschutz Medical Campus, University of Colorado, Aurora, CO 80045, USA; 4Department of Pharmacology, Anschutz Medical Campus, University of Colorado, Aurora, CO 80045, USA

**Keywords:** extracellular vesicles, hemorrhagic stroke, inflammation, Glasgow Outcome Scale, Glasgow Coma Scale-Extended, cytokines, chemotaxis

## Abstract

Background: A major contributor to disability after hemorrhagic stroke is secondary brain damage induced by the inflammatory response. Following stroke, global increases in numerous cytokines—many associated with worse outcomes—occur within the brain, cerebrospinal fluid, and peripheral blood. Extracellular vesicles (EVs) may traffic inflammatory cytokines from damaged tissue within the brain, as well as peripheral sources, across the blood–brain barrier, and they may be a critical component of post-stroke neuroinflammatory signaling. Methods: We performed a comprehensive analysis of cytokine concentrations bound to plasma EV surfaces and/or sequestered within the vesicles themselves. These concentrations were correlated to patient acute neurological condition by the Glasgow Coma Scale (GCS) and to chronic, long-term outcome via the Glasgow Outcome Scale-Extended (GOS-E). Results: Pro-inflammatory cytokines detected from plasma EVs were correlated to worse outcomes in hemorrhagic stroke patients. Anti-inflammatory cytokines detected within EVs were still correlated to poor outcomes despite their putative neuroprotective properties. Inflammatory cytokines macrophage-derived chemokine (MDC/CCL2), colony stimulating factor 1 (CSF1), interleukin 7 (IL7), and monokine induced by gamma interferon (MIG/CXCL9) were significantly correlated to both negative GCS and GOS-E when bound to plasma EV membranes. Conclusions: These findings correlate plasma-derived EV cytokine content with detrimental outcomes after stroke, highlighting the potential for EVs to provide cytokines with a means of long-range delivery of inflammatory signals that perpetuate neuroinflammation after stroke, thus hindering recovery.

## 1. Introduction 

Cerebrovascular accident (CVA), or stroke, is the worldwide leading cause of long-term disability, occurring at any age and across all demographics [1]. Hemorrhagic stroke often results in poor prognosis with a disturbing mortality rate of ~74%. Of those who survive the initial stroke event, 90% suffer from a permanent lifelong disability [1] and an estimated financial burden of 34 billion USD. The incidence of stroke continues to rise, even in younger age groups, with nearly half of the US population predicted to have cerebrovascular disease in some form by the mid 2030s [2]. The morbidity and mortality of hemorrhagic stroke, in addition to the extreme financial burden it imposes, highlight the need for better prognostics to identify patients in need of early interventions.

A major contributor to disability after stroke is secondary brain damage induced by the inflammatory response [3]. After stroke, concentrations of a wide range of cytokines increase globally within the brain, cerebrospinal fluid, and in peripheral blood/plasma [4,5,6], several of which have been correlated to worse outcomes after stroke [7,8]. Based on these findings, clinical trials are currently in progress to test the efficacy of inhibition of cytokine-mediated inflammatory processes. Inflammatory regulators, such as the IL1 system [9], are known mediators of inflammation in stroke, and this has been targeted in several clinical trials [10]. Administration of IL1 receptor antagonists within several hours after CVA improves overall patient outcome compared to placebo-treated patients [11]. This highlights the pathways of cytokine-mediated inflammation as promising avenues for stroke treatment. However, a comprehensive study investigating cytokine profiles after hemorrhagic stroke is needed to identify the most efficacious treatments as well as to develop a biomarker panel for the identification of patients at risk of cytokine-mediated neuroinflammation.

Cytokine diffusion, and thus their signaling influences, are functions of cellular release, cellular uptake (via specific cytokine receptors), and extracellular degradation [12,13,14]. An alternative means of cytokine release and distribution could come in the form of packaging and enrichment in extracellular vesicles (EVs), which could provide a means of protected transport. EVs are nano-sized, membrane-enclosed vesicles released extracellularly from essentially all cell types and into all biofluids [15]. They may derive from the endosomal system (referred to as exosomes) or may bud directly from the plasma membrane (sometimes called ectosomes or microvesicles) [16], as well as other forms of secreted vesicles [17]. EVs released into extracellular fluids contain and can transfer DNA, various RNAs, lipids, and proteins [18] including cytokines for cell–cell communication, and they possess the capability of crossing the blood–brain barrier [19]. Cytokine-enriched EVs would offer a focused profile of the post-stroke cytokine response from a minimally invasive, peripheral source in circulating plasma. This avoids retrieval of CSF through painful lumbar puncture for similar analysis, making EVs ideal targets for the identification of diagnostic biomarkers.

In this study, we quantified the abundance of 80 cytokines in the plasma of hemorrhagic stroke patients. Further, we quantified the same 80 cytokines on patient plasma EVs, assessing both EV membrane-bound and internally sequestered cytokines. We found EV-associated cytokines to correspond to worse patient outcome compared to cytokine profiles from plasma alone. We report here that EV-associated cytokine profiles are largely correlated to worse acute (GCS) and chronic (GOS-E) neurological conditions regardless of the cytokines′ individual predicted effects on neuroinflammation or neuroprotection.

## 2. Results

### 2.1. Patient Demographics and Information

In total, 31 patients were included in this analysis (21 male and 10 female). Four patients were African American, 3 were Hispanic, and 24 were Caucasian. The average age was 51.17 ± 15.88 SD. Age was not significantly correlated to either GCS or GOS-E (*p* = 0.66, *r* = 0.08; *p* = 0.49, *r* = −0.13, respectively). The average BMI was 25.8 ± 4.3 SD, which was not correlated to either GCS or GOS-E (*p* = 0.75, *r* = 0.06; *p* = 0.77, *r* = −0.05, respectively). Multiple types of hemorrhage were recorded, including subarachnoid hemorrhage (SAH) (*n* = 7); subdural hemorrhage (SDH) (*n* = 6); intracranial hemorrhage (ICH) (*n* = 3); intraparenchymal hemorrhage (IPH) (*n* = 4); intraventricular hemorrhage (IVH) (*n* = 3); and a combination of up to 3 hemorrhage types (*n* = 8). Upon admission, the average GCS score was 10.54 overall, 10.62 ± 4.33 SD for males, 10.40 ± 2.91 SD for females, 13.75 ± 0.96 SD for IPH, 7.60 ± 3.27 SD for IVH, 9.40 ± 4.30 SD for ICH, 11.44 ± 4.42 SD for SAH, and 9.30 ± 3.34 SD for SDH. The average hemorrhage volume was 32,855.49 mm^3^ ± 24,403.33 SD, which was negatively correlated to the GCS (*p* = 0.012, *r* =−0.45) and not significantly correlated to the GOS-E (*p* = 0.88, *r* = −0.03). The type of hemorrhagic stroke was not observed to affect GCS or GOS-E. Healthy subjects include 8 males and 6 females averaging 53 years old ± 13.42 SD. A healthy pooled plasma sample was also included to provide an “averaged” cytokine level for healthy plasma. Pooled healthy donor plasma averaged 52 years old ± 10.96 SD. These and other demographic data, as well as outcome scores, are shown in Appendix A. Appendix A breaks down the GCS [20] and GOS-E [21,22] scores by clinical criteria to provide a context for these scores in terms of patient impairment.

### 2.2. A Limited Number of Plasma Cytokines Are Associated with Positive and Negative GCS Scores and Positive GOS-E Scores in Hemorrhagic Stroke Patients

We evaluated the relationships among global plasma cytokine profiles with respect to GCS (early phase measure) and GOS-E scores (30+ day outcome measure). We correlated the concentrations of these plasma cytokines to patient GCS, recorded at the time of blood draw—5–7 days after admission—and to their GOS-E scores obtained at least 3 months after stroke; these were then compared to plasma cytokine concentrations of the healthy controls. Of the cytokines correlated with GCS from the blood plasma only, granulocyte chemotactic protein 2 (GCP-2/CXCL6) was shown to correlate to the most negative GCS (*p* = 0.04, *r* = −0.340), followed by Eotaxin-3/CCL26 (*p* = 0.057, *r* = −0.421) and hepatocyte growth factor (HGF) (*p* = 0.046, *r* = −0.420). Insulin-like growth factor binding protein-4 (IGFBP-4) (*p* = 0.067, *r* = −0.387) was correlated to a more negative GCS score, and tissue inhibitor of metalloproteinases 2 (TIMP-2) (*p* = 0.063, *r* = 0.393) was correlated to a more positive GCS score. Cytokines considered to be inflammatory (GCP-2, Eotaxin-3) were correlated to more negative GCS than those considered neuroprotective (HGF, IGFBP-4, TIMP-2) (Figure 1).

Four cytokines from plasma alone that correlated to GOS-E show positive regression values. Angiogenin (*p* = 0.0494, *r* = 0.414), RANTES/CCL5 (*p* = 0.043, *r* = 0.426), neutrophil-activating peptide *2* (NAP-2/CXCL7) (*p* = 0.055, *r* = 0.405), and osteopontin/SPP1 (*p* = 0.0508, *r* = 0.412) were correlated to positive GOS-E scores (Figure 2). RANTES, a cytokine with a mixed inflammatory and neuroprotective profile [23,24], was most strongly correlated to GOS-E, whereas neuroprotective cytokines NAP-2, osteopontin, and angiogenin were correlated slightly less strongly to GOS-E. 

### 2.3. EV Surface-Bound Cytokines Correlated to Negative GCS and GOS-E Scores

In different systems, including blood, cytokines appear to be on EV surfaces as well as contained inside them [19,25,26]. Thus, we also computed the statistical regression of presumably EV membrane-bound cytokines (i.e., from intact EVs) to patient outcome scores and compared them to cytokine concentrations on the surfaces of the plasma EVs from healthy controls. Six anti-inflammatory and nine inflammatory cytokines were negatively correlated to GCS scores. The anti-inflammatory cytokines most correlated to negative GCS at 5–7 days post stroke include IGF-1 (*p* = 0.022, *r* = −0.409), HGF (*p* = 0.031, *r* = −0.388), and EGF (*p* = 0.033, *r* = −0.384). Of the inflammatory cytokines identified, the highest correlations to negative GCS include macrophage-derived chemokine (MDC/CCL22) (*p* = 0.016, *r* = −0.432), interleukin 7 (IL7) (*p* = 0.02, *r* = −0.415), and GRO/CXCL1 (*p* = 0.027, *r* = −0.398) (Figure 3). Unlike cytokines correlated to GCS in plasma, the EV surface-bound cytokines do not follow a regression trend based on their pro- or anti-inflammatory properties.

We identified 10 anti-inflammatory and 14 inflammatory cytokines from EV surfaces that correlated to GOS-E. The chemokines MCP-4/CCL13 (*p* = 0.03, *r* = −0.389) and fractalkine/CX3CL1 (*p* = 0.0319, *r* = −0.386) were shown to correlate to the most negative regression values concerning GOS-E outcome score, followed by growth factor PlGF/PGF (*p* = 0.038, *r* = −0.376) (Figure 4). EV surface-associated cytokines compared to GOS-E score again did not follow any regression trend based on predicted pro- or anti-inflammatory properties.

### 2.4. Intra-EV Cytokines Are Mostly Correlated with Negative GOS-E Scores

Plasma EVs were stripped of surface proteins by protease K treatment, and the EVs themselves were re-isolated by ultracentrifugation, followed by lysis to release intra-EV cytokines. The only intra-EV cytokine detected to correlate to GCS was insulin-like growth factor-binding protein 2 (IGFBP-2) (*p* = 0.086, *r* = −0.313), which indicated a negative GCS (not shown). Four anti-inflammatory intra-EV cytokines were correlated to GOS-E outcomes, including fibroblast growth factor 6 (FGF6) (*p* = 0.036, *r* = −0.379), oncostatin M (*p* = 0.076, *r* = −0.323), stem cell factor (SCF/KITLG) (*p* = 0.053, *r* = −0.351), and vascular endothelial growth factor (VEGF) (*p* = 0.033, *r* = −0.383) (Figure 5). The cytokine oncostatin M may have inflammatory characteristics but has recently been shown to exhibit neuroprotective effects after ischemic stroke [27,28]. The inflammatory cytokine MCP-1/CCL2 (*p* = 0.003, *r* = −0.516) was identified as having the most negative affect on GOS-E, followed by IL15 (*p* = 0.006, *r* = −0.484), RANTES/CCL5 (*p* = 0.006, *r* = −0.483), and IFN-γ (*p* = 0.008, *r* = −0.468). Inflammatory cytokine MIP-3 alpha/CCL20 (*p* = 0.099, *r* = 0.302) was the only cytokine correlated to a positive GOS-E regression score (Figure 5).

### 2.5. EV-Associated Cytokines Share Protein–Protein Inflammatory Pathways

The Metascape platform provides insight and understanding of common and unique protein–protein interactions and pathways achieved by the analysis of target proteins against 40 independent “omic” databases [29]. This platform was used to help understand potential common pathways and mechanisms shared by the cytokines we identified and to group them into gene ontology-type presentations. Enrichment analysis of both GCS- and GOS-E-correlated EV cytokines revealed common pathways and enrichment clusters involved in insulin-like growth factor signaling, inflammation, and pathogenic immune responses. For this analysis, we used various groups of cytokines (EV membrane bound, or internally sequestered within EVs) with correlations to GCS, GOS-E, or both. Using EV surface cytokines and intra-EV cytokines that correlated to GCS, the analysis identified the interactions with, and regulation of, the insulin-like growth factor receptor signaling pathway (Log10 (*P*) = −2.2) (Figure 6). No additional interactions were elucidated with the inclusion of identified plasma cytokines correlated to GCS. The analysis for membrane-bound and intra-vesicle cytokines correlated to GOS-E showed interactions with the chronic inflammatory response (Log10 (*P*) = −2.9) and its regulation (Log10 (*P*) = −3.0) (Figure 6), along with pain sensation. When plasma cytokines that were correlated to GOS-E were added to the analysis, they revealed no additional interactions. We identified four EV membrane surface-bound cytokines, MDC/CCL2, M-CSF/CSF1, IL7, and MIG/CXCL9, that had significant correlations to negative scores in both GCS and GOS-E (Figure 1 and Figure 2). These four cytokines were significantly linked to enrichment clusters involving both influenza A (Log10 (*P*) = −2.25) and herpes simplex infection (Log10 (*P*) = −3.85). The regulation of phagocytosis (Log10 (*P*) = −3.3) was also correlated. No cytokines identified from plasma alone or from lysed vesicles were shown to overlap between GCS and GOS-E. The resultant protein networks and enrichment clusters we found would be influenced by our choice to use cytokines with correlations up to *p* = 0.1 and their correlations to GCS or GOS-E; any bias would be assuaged by using the cytokines that did not correlate to GCS/GOS-E for background subtraction.

### 2.6. Plasma and EV-Associated Cytokines Predict Differential Changes in Immune Cell Movement and Function

To assess potential cellular implications of cytokine levels in the plasma compartment versus the different EV compartments, we employed Core Analyses and Comparison Analyses in the Ingenuity Pathway Analysis (IPA). The cytokine array scores from each patient or healthy donor from plasma, EV lumen, and EV surface cytokines were quantitatively analyzed to generate log(2) fold change data of stroke patient values compared to healthy donor values. The algorithm generated heat maps from activation z-scores listing the top 50 diseases and biofunctions (Figure 7), where we note a preponderance of categories regarding myeloid cell and lymphocyte movement/migration, homing, proliferation, induction, and activation. EV-associated cytokines (both EV lumen and EV surface) generally have higher activation z-scores than plasma cytokines and would be affiliated with stroke patient cytokine scores based on the nature of the comparison.

Using IPA-extracted pathway data, in Figure 8A,B we show the top two derived networks (out of 14); network scores ranged from 25 down to 1. Scores are derived from Fisher′s exact test and are −log (*p* values) indicating the probabilities of random protein associations. Both networks or interactomes had scores of 25, and each had 13 focus molecules or nodes from which the networks initiate. Higher cytokine scores are indicated by red shading and lower values by green shading, which are again relative to their representation in stroke patients vs. healthy donors.

We note in Figure 8A, Network 1 (“Cellular Movement; Hematological System Development & Function; Immune Cell Trafficking”), the TNFA node and inflammatory signaling axis—IL1/IL6/TNF—and connections to the TNF receptor and lymphotoxin (TNFB/LTA) [30]. The constituent cytokine/receptor levels change across compartments (plasma vs. EV lumen vs. EV surface), suggesting a possible differential effect on cytokine recipient cells depending on the plasma/plasma component source of the cytokines. Of the CC chemokine ligands, several remain similarly elevated across the compartments (CCL13, CCL8, CCL7, and CCL6), while others are variable (CCL26/Eotaxin-3, CCL24/Eotaxin-2, CCL18, and CCL15). Of the CXC chemokines, CXCL6 and CXCL17 remain elevated across compartments, while PPBP/NAP-2/CXCL7 and CXCL5 vary.

Network 2 (Figure 8B, “Cell Signaling; Cell-Cell Signaling & Interaction; Cellular Growth & Proliferation”) contains numerous interleukin family members and a variety of growth factors. IL1B and the type I interferon IFNB display relatively stable and high expression across the compartments, while the other interleukins vary. Similarly, of the growth factors, CSF2/GM-CSF, LIF, and HGF/SF show high expression across the compartments, while the rest vary. Curiously, all but TGFB are lowest in the plasma and higher in the EV compartments. CXCL10/IP-10 varied as well. The network is very dense, with the interactome showing many connections and interactions amongst the nodes. The two networks further implicate cytokines and growth factors in immune cell signaling, trafficking, and migration, all of which could be expected following a stroke incident.

## 3. Discussion

In this study, we investigated the concentrations of 80 cytokines present in circulating plasma or associated with extracellular vesicles of hemorrhagic stroke patients approximately 1 week after a stroke event. We demonstrate that 31 hemorrhagic stroke patients spanning five stroke etiologies may be statistically relegated to show a correlation between EV-associated cytokine concentrations and patient acute conditions and chronic neurological outcome scores. The correlative cytokines differ in regression effect and significance depending on their detection in the plasma or association with EVs, suggesting that potential sequestration or delivery of cytokines by EVs is detrimental to recovery after hemorrhagic stroke across all etiologies. The strengths of the associations of the correlation coefficients ranged from medium in most cases to strong in a few cases. As these were regression analyses, we used the full range of GCS and GOS-E scores available to elicit outcome score trends correlating to cytokine levels (and their plasma vs. EV fractions). We did not define cutoff values to make specific determinations, as this may require subjective interpretation of individual patients’ clinical status, particularly for longer term outcomes. This would be an ideal area of future investigation.

Extracellular RNAs, such as microRNAs, presumably from plasma EVs, have been proposed to distinguish ischemic from hemorrhagic stroke [31]. In general, EVs are recognized as potential biomarkers for stroke but are more frequently studied as putative therapeutics [32]. Our efforts here concerning cytokine-laden EVs as biomarkers for CVA outcome appear to be novel.

A recent investigation by Fitzgerald et al. [19] concluded that cytokines associated with EVs populate both inside and outside the vesicle membrane; cytokine surface binding to the external membrane may be in the context of receptors [25], and, by other mechanisms, cytokines may be packaged within the vesicle lumen itself [19]. Due to sequestration within vesicles, these encapsulated cytokines may not be detected in standard cytokine assays [19]. These observations highlight two potential therapeutic avenues for hemorrhagic stroke. First, EVs were shown to be enriched with cytokines both on the membrane surface and sequestered within. This implicates EVs as a concentrated and protected source of cytokines not previously considered when searching for cytokine biomarkers related to CVA. Cytokine biomarkers (and numerous others) from both blood and cerebrospinal fluid (CSF) have been suggested for early neurological degeneration in acute stroke settings [33] along with other early (acute ischemic stroke setting) markers for longer term outcome [34]. Other studies in acute ischemic stroke patients implicate serum/plasma cytokines relative to infarct size, to stroke severity (measured by the NIHSS), and to short-term prognosis (outcome at 3 months determined by the mRS) [35]. In general, there are fewer studies involving hemorrhagic stroke, but the relationships with pro- and anti-inflammatory cytokines in blood and CSF and patient outcomes tend to be defined by the interplay between macrophage and T cells [36,37].

The use of plasma EVs as reservoirs of predictive biomarkers for stroke outcome provides a cytokine-rich diagnostic source that is relatively easy to retrieve and minimally invasive compared to the spinal tap for CSF. Secondly, EV sequestration of cytokines may veil the true extent of their influence in post-stroke neuroinflammation. Failure to detect encapsulated cytokines through traditional assays leaves a substantial gap in the understanding of how neuroinflammation may be perpetuated, affecting CVA recovery. Curiously, we have shown similar correlations to negative outcomes in both pro-inflammatory and anti-inflammatory cytokines; however, the populations of inflammatory cytokines correlated to a poor outcome outnumber non-inflammatory cytokines when measured as bound to the surface of the extracellular vesicles as well as protected within. The impacts of cytokine localization in/on EVs may have profound influences on recipient cells, which may be determined by cytokine receptors on those cells, or modes of internalization and intracellular release of EV luminal cytokines. While this topic goes beyond the coverage of this study, further research is clearly needed to assess the potentially differential effects of EV cytokine localization both at the level of recipient cells and in overall systemic impacts.

Diagnostically, the true inflammatory nature of these EVs may not be understood through traditional cytokine measurements of EVs, and other methods may need to be employed. The Metascape platform was used to gain insight into the interactions of cytokines identified as significant or trending toward significant in correlation to GCS and GOS-E. Among the 80 cytokines measured, those that were not correlated significantly or trending toward significance were used as background subtraction in an effort to detect cytokine interactions and biological pathways mediated only by those cytokines correlated to GCS or GOS-E. For instance, insulin-like growth factor signaling is well recognized in the context of ischemic stroke [38], suggesting the early stages of neuroprotection in hemorrhagic stroke. On the other hand, for CVAs in general, the role of inflammation, both acute and chronic, remains complicated [39] and dovetails into cytokine-driven immune mechanisms and arachidonic acid metabolites [40] (e.g., “Cellular response to organic cyclic compound”). The pathways identified through Metascape highlight the complexity of cytokine interaction and their downstream effects, while also providing potential direction for more focused study. Further investigation is needed to evaluate whether the pro-inflammatory effect of EV-associated cytokines overcomes any beneficial effects of anti-inflammatory cytokines packaged in the same vesicle. These observations reveal the potential for great insight into post-stroke neuroinflammation from an easily accessible source not previously considered.

GCS has been shown to correlate to overall patient outcome after brain injury [41]. We identified four EV membrane surface-bound cytokines, MDC/CCL2, M-CSF/CSF1, IL7, and MIG/CXCL9, whose concentrations all have significant correlation to a negative score in both GCS and GOS-E (Figure 1 and Figure 2). All four of these cytokines have been shown to contribute to the polarization of TH1 or TH2 T-helper cells [35,42,43,44,45]. The commonality in their roles suggests that polarization in T-helper cells may be a driving factor in the sustained neuroinflammatory response continued from the acute neurological condition (GCS) to overall patient outcomes reported by GOS-E.

A critical part of the immune response is the polarization of multipotential T-cells, restricting them to mature into T-helper 1 (TH1) cells, which are responsible for inflammatory responses to infection, or T-helper 2 (TH2) cells, which aid in adaptive immunity and are considered neuroprotective [44,46]. After polarization, TH1 and TH2 cells secrete the hallmark cytokines IFN-γ and IL4, respectively [44]. When these concentrations were compared to healthy controls, IL4 and IFN-γ levels detected in our three sample preparations largely varied among both groups. However, EV membrane-bound IL4 levels were significantly correlated with stroke compared to healthy subjects (*p* = 0.038) (Appendix A). This correlation further implicates EV surface-bound cytokines as drivers of EV-to-cell inflammatory signaling in both acute (GCS) and chronic (GOS-E) neurological conditions after hemorrhagic stroke. This is intriguing in light of the neuroprotective roles for TH2-type cytokines and monocyte interactions believed necessary for neural repair following stroke [47,48].

The involvement of lymphocytes and myeloid/monocytic cells in terms of development and migration is also implied by the IPA data (Figure 7 and Figure 8). We note the roles of CCLs-1, -5, -8, -13, and -18 in T cell responses in immune function, particularly in trafficking and interactions between lymphocytes and monocyte-derived cells [49]. We highlight a direct connection between the transcription factor AP1/JUN and IL1B in Figure 8B; in a murine ischemia/reperfusion stroke model, downregulation of AP1 increases IL1B in association with brain injury [50]. Of course, IL1B (and other IL1 family members) are heavily documented in stroke pathology and treatment [10]. Factors such as TGFB and VEGFA are prominent nodes (Figure 8B), and the former connects with SMAD2/3. These are all involved in vascular remodeling and blood–brain barrier repair via monocytes/macrophage in hemorrhagic transformation following ischemic stroke [51]. The interplay of these immune cell types and EVs in stroke certainly warrants further study.

Tying into the IPA data, the chemokine RANTES/CCL5 was identified to significantly correlate to patient outcomes in both patient plasma and within EVs. Interestingly, the concentrations of RANTES were correlated to a more positive outcome as a free soluble form in the plasma and a more negative outcome within the EV lumen (Figure 2 and Figure 5). RANTES is widely recognized as an inflammatory cytokine with a major role in the recruitment of leukocytes to inflammatory sites [52] but with complicated influences, particularly regarding its receptor, CCR5 [53,54]. However, recent studies suggest that RANTES may also contribute to neuroprotective processes. Tokami et al. [24] demonstrated that increased expression of RANTES after ischemic stroke also increases the upregulation of growth factors brain-derived neurotrophic factor (BDNF), epithelial growth factor (EGF), and VEGF in neuronal cells [24]. Within blood plasma, our study also detected concentrations of EGF, VEGF, and BDNF to correlate to a more positive outcome (Appendix A); however, these were not statistically significant. Concentrations of BDNF and EGF within the EVs were correlated significantly to a more negative patient outcome. Further investigation is required, though initial data supports the findings that RANTES may have both neuroinflammatory and neuroprotective roles after brain injury and suggest that the differing effects of RANTES may also rely on whether it is delivered via free form within plasma or sequestered within EVs.

Of all cytokines profiled, four of the five most negatively correlated to outcome were cytokines sequestered within EVs: MCP-1/CCL2 (*p* = 0.003, *r* = −0.516), IL15 (*p* = 0.006, *r* = −0.484), RANTES/CCL5 (*p* = 0.006, *r* = −0.483), and IFN-γ (*p* = 0.008, *r* = −0.468) (Figure 5). Each cytokine has a substantial contribution to the immune response, including the recruitment of monocytes, neutrophils, and lymphocytes (MCP-1, RANTES) [52,55]; the activation of T-cells (IL15) [56]; and the activation of macrophages (IFN-γ) [57]. These data support our original hypothesis that encapsulated cytokines within EVs may be a source of inflammatory signaling negatively affecting patient recovery. Future therapeutic research may benefit from further investigation of EV-delivered cytokines. We point out that our study utilized blood/EVs at 5–7 days following the stroke event; this is a critical time period after the secondary disruption/dysfunction of the blood–brain barrier [58]. There may be an increased release of EVs from brain tissue that can now access the peripheral blood (and vice versa). Further work is necessary to determine the origins of the cytokine-laden EVs as well as their impacts on recipient cells. Indeed, EVs are speculated to mediate cross-talk between the brain and immune responses in brain injuries such as stroke, particularly in an inflammatory context [59].

There are certainly caveats associated with this work. One assumption made is that EVs are inherently stable and are not spontaneously lysing to release internalized cytokines. Previous work [19] demonstrated stable, non-degraded cytokines within EVs collected from various sources, suggesting that our collected plasma EVs are likely stable as well. That same study also showed using “spike-in” assays that cytokines do not seem to adhere to EVs non-specifically. We also assume that protease K treatment effectively digests essentially all EV surface proteins, at least to the point of epitope loss for antibody recognition. This is considered to be a very effective proteolysis method concerning EV surfaces [60], and the supernatants from the protease K-treated EVs that are pelleted by ultracentrifugation do not show reactivity on the cytokine array (Appendix A). Nonetheless, we cannot rule out that some cytokine epitopes may be preserved by EV lipids or EV surface carbohydrates and would thus be counted as the “internal” pool of EV cytokines. However, this does not change the overall outcomes regarding EV-based possession and the transport of cytokines impacting patient outcomes after CVA. Another concern may be the presence of lipoprotein particles that co-precipitate with the ExoQuick Ultra protocol. We evaluated this by Western blotting for the apolipoproteins APOB and APOE (Appendix A) and saw both of these apolipoproteins in selected EV preparations. This finding is similar to that of Brennan et al. [61] (which found APOB and APOE in serum EV preparations using a variety of methods, including ExoQuick Plus). We point out that in APOB and APOE, preparations from many different cell types are listed as components of EV in both the Vesiclepedia (http://www.microvesicles.org/, accessed on 20 May 2021) and ExoCarta (http://www.exocarta.org/, accessed on 20 May 2021) databases. Our own work identified APOA1 and APOA4 in the proteomic analyses of EVs derived from human medulloblastoma cell lines and murine glioma cells [62,63], with APOB and APOE identified in EVs from other tumor cell lines (Graner, unpublished data). Further, these proteins are thought to be components of the “protein corona” that accumulates on serum- or plasma-treated nanoparticles [64,65]. While we cannot rule out that lipoprotein complexes may co-precipitate in our plasma EV preparations, there remains a formal possibility that apolipoproteins may be true EV constituents, and that debate will not be settled here. Further, it is unclear that lipoprotein complexes contain cytokines, or to what degree, and, thus, any role of lipoprotein particle-associated cytokines remains unknown.

We also note that the relatively small number of patients in the study precludes a robust multivariable analysis, particularly with the number of variables we already describe, along with many that we do not. For instance, individualized treatments, ranging from aspirin to surgical intervention, were not accounted for in this study (and we did not necessarily have access to such information). A detailed study involving the effects of anti-inflammatory agents or other clinical mediation in plasma/EV cytokine profiles and patient outcomes is beyond the scope of this work but would be a worthy effort.

Edema is another clinical aspect of hemorrhagic stroke with important clinical implications [66] and a close association with inflammatory processes [67]. Edema, particularly perihematomal edema, is dynamic and rather variable over the 2 weeks following ICH onset [68]. Edema can also be challenging to image [69,70]; for these reasons, we chose only to use hemorrhage volume at the earliest available time point in our data and evaluations. However, the relationships between this ICH sequela and biofluid EV cytokine changes would surely be a valuable future study, as the temporal impacts on inflammation may hold clues about the pathophysiology of stroke and also potentially targetable areas of intervention.

Further work is necessary to determine the cellular/tissue recipients of EV-delivered cytokines, as these will likely be points of therapeutic interventions. The effects of cytokine presence on EVs as well as within EVs is a new area of research; in conjunction with free plasma cytokines, there is great potential to alter the landscape of inflammatory responses during and following CVA. We point out that this is not a definitive biomarker study, given the complexity of the study, but it provides a novel springboard for future work.

## 4. Materials and Methods

### 4.1. Inclusion Criteria

Study subjects were admitted to the University of Colorado Hospital within 8 h after onset of a stroke event. Thirty-one hemorrhagic stroke patients and 15 healthy non-stroke subjects were included in the study. This was the first stroke event for all stroke subjects, and the course of clinical care was similar across all stroke subjects as per standard of care suggested by AHA/ASA guidelines [71]. All experiments and collections of patient samples and data were authorized by the Colorado Multiple Institutional Review Board (COMIRB #13-2605). Patient and healthy donor demographics are available in Appendix A.

### 4.2. Peripheral Blood Collection

Whole blood was collected from the patients at 5–7 days after the stroke event. Ten milliliters of whole blood were collected in EDTA (K2) blood tubes (Becton Dickinson, GA, USA, BD366643). Whole blood was centrifuged at 1850 RPM (780× *g*) (Beckman Coulter, IN, USA, Allegra 6) for 10 min, and plasma was collected. Plasma was collected from healthy subjects (8 males and 6 females, avg 53 years old) as well as a pooled healthy donor plasma sample (Innovative Research, MI, USA, ISERAB-100 mL). Plasma was stored at −80 °C until analyzed.

### 4.3. Volumetric Analysis of Brain Hemorrhagea

Patient non-contrast head computed tomography (CT) was performed within 12 h of hemorrhage and prior to surgical intervention per standard of care. Volumetric analysis of CT images was conducted through manual segmentation of the visualized edema region, using either BrainLab (BrainLab AG, Munich, Germany) or Insight Toolkit (ITK)-SNAP (ver. 3.80) [72].

### 4.4. Isolation of Extracellular Vesicles

In total, 250 μL of plasma was used with the ExoQuick^®^ Ultra EV Isolation Kit for Serum and Plasma (System Biosciences, CA, USA, EQULTRA-20A-1) for the isolation of EVs by precipitation per kit instructions. This includes an initial centrifugation of 3000× *g* for 15 min for further removal of platelets. We utilized ExoQuick^®^ due to the small volumes of materials available as in [19], and we are aware of the issues involving its use, particularly from complex biofluids such as plasma [73]. Note below that we do use ultracentrifugation for secondary isolation of surface protein-stripped EVs. Total EV protein concentration was measured by a bicinchoninic acid (BCA) assay (Thermo Scientific, IL, USA, A53225).

Regarding the presence of platelets in the plasma, following the 780× *g* centrifugation of the whole blood to collect plasma, per ExoQuick^®^ instructions the plasma was spun at 3000× *g* for 15 min; this step should remove platelets. Further, in studies where cytokines associated with platelet EVs were measured, the vesicles were from isolated platelets that were activated to release larger particles (frequently termed “microparticles” [26,74,75]). As we did not isolate platelets, nor activate them, and our EVs were of considerably smaller diameters (Appendix A), we rule out significant contributions to EV cytokine profiles from platelets.

### 4.5. Validation of Extracellular Vesicle Isolation

EVs isolated from 2 healthy donor individuals and from two CVA patients were assessed for EV protein markers using ExoCheck™ arrays (System Biosciences, CA, USA, EXORAY210A-8) (Appendix A). Additionally, isolated extracellular vesicles from all patients and healthy donors were analyzed for size through the nCS1TM Nanoparticle Analyzer System (Spectradyne, CA, USA) for verification of EV sizes and concentrations (Appendix A).

Further characterization of extracellular vesicles was performed through transmission electron microscopy (Appendix A). For each sample, 8 µL was applied to a copper mesh grid coated with formvar and carbon (Electron Microscopy Sciences, PA, USA) for 2 min and then gently blotted off with a piece of Whatman filter paper. The grids were rinsed with transfers between 2 drops of MilliQ water, blotting between each transfer. Finally, the grids were stained with 2 drops of a 0.75% uranyl formate solution—a quick rinse with the first drop followed by 20 s of staining in the second drop. After blotting, the grids were allowed to dry for at least 10 min. Samples were imaged on an FEI Tecnai G2 Biotwin TEM (ThermoFisher, OR, USA) at 80 kV with an AMT side-mount digital camera (AMT Imaging, MA, USA).

### 4.6. Surface Protein Stripping and Lysis of Extracellular Vesicles

Whole EVs isolated through Exoquick^®^ Ultra were stripped of surface proteins with 2 mg/mL proteinase K (New England Biolabs, MA, USA, P8107S) in NEB buffer 2 (New England Biolabs, MA, USA, B7202S) and 50 mg/mL of BSA (Fisher Scientific, MA, USA, 23209) for 1 h at 37 °C with gentle shaking. Whole EVs were then ultra-centrifuged at 150,000× *g* in a Beckman Ti 70.1 rotor (*k* factor = 150) for 2 h at 4 °C (adapted from methods from Smyth et al. [76]) to separate EVs from stripped surface proteins. The supernatant (containing stripped and proteolyzed cytokines) was discarded. A total of 250 μL of 1x RIPA buffer (Fisher Scientific, MA, USA, 89900) with Triton X-114 (Sigma Aldrich, MO, USA, X114–100 mL), plus protease and phosphatase inhibitors (Roche, Mannheim, GER, 04-906-837-00, 04-693-124-00), were used to lyse vesicles with sonication (Fisher Scientific, MA, USA, FS30) performed for 1 h at 4 °C. The total protein concentration of lysed EVs was measured by BCA assay.

### 4.7. Plasma, EV Surface-Bound, and Intra-EV Cytokine Quantification

Cytokines in plasma, cytokines bound to membranes of isolated unlysed extracellular vesicles, and cytokines packaged within the isolated vesicles were measured. The total protein concentrations of these 3 preparations from subject samples were quantified using BCA assay and adjusted to 25 μg/mL for cytokine quantification via array measuring abundance of 80 cytokines (Raybiotech, GA, USA, AAH-CYT-5-8). Samples were analyzed per the manufacturer’s instructions with primary antibody cocktail probing performed overnight at 4 °C and secondary antibody probing for 2 h at room temperature. The final cytokine arrays were imaged with a Cell Biosciences FluorChemQ Multiimage III chemiluminescence imaging cabinet (FluorChemQ, ProteinSimple, CA, USA). Fluorchem imaging software was used to quantify array signal and normalized to internal controls included on the array. Quantification results for cytokines from the different plasma/EV fractions are found in Appendix A.

### 4.8. Detection of Apolipoproteins

The presence of apolipoprotein B (APOB) and apolipoprotein E (APOE) was evaluated by Western blot, adapted from Brennan et al. 2020 [29] (Appendix A). After lysis and quantification of extracellular vesicles as described above, protein (4  µg) was mixed with 2X Laemmli buffer with 80 mM DTT (Biorad, Hercules, CA, USA, 1610737) and heated to 95  °C for 10  min. Protein was added to wells of fixed-percentage (12%) Tris-Glycine gels (Biorad, 5671043). Gels were run in Tris-Glycine-SDS running buffer (Biorad, 1610732) at 200  V for 43  min. Proteins were then transferred to 0.45  µm nitrocellulose membrane using an iBlot™ 2 blotting system (Fisher Scientific, MA, USA, IB21001) with variable voltages of 20 V for 1 min, 23 V for 4 min, and 25 V for 2 min for a total of 7 minutes. The membranes were blocked for 1 h at RT in 1X TBS containing 5% (*w*/*v*) bovine serum albumin. Proteins were detected by incubation with primary antibodies against APOE (R&D Systems, AF4144, 1 µg/mL) and APOB (R&D Systems AF3556, 1 µg/mL) in blocking solution overnight at 4  °C. Following 3 × 5 min washes of TBS with 0.1% Tween-20 (TBST), membranes were incubated in HRP-labeled rabbit anti-goat IgG secondary antibody (R&D Systems HAF017) diluted 1:1000 in blocking solution for 1 h at RT. The blots were washed 5 times in TBST for 5 min and washed a final time in TBS for 5 min. Blots were developed with SuperSignal™ West Pico PLUS Chemiluminescent Substrate (ThermoFisher 34579) and were imaged with a Cell Biosciences FluorChemQ Multiimage III.

### 4.9. Outcome Reporting

Patient physical and neurological status was evaluated by various stroke etiology-dependent scales, including the Glasgow Coma Scale (GCS) [41], National Institute of Health Stroke Scale (NIHSS) [77], Hunt and Hess score [78], and Fisher scale [79] for subarachnoid hemorrhage (SAH), and intracerebral hemorrhage score [80] on admission by a member of the patient’s care team. At hospital discharge, the modified Rankin Scale (mRS) [81] was used to evaluate patient outcomes (Appendix A). After at least 3 months post-stroke event, the Glasgow Outcome Scale-Extended (GOS-E) [82] was applied to gauge patient outcome.

### 4.10. Metascape Analysis

Protein–protein interactions between all cytokines that correlated to GCS or GOS-E outcome up to *p* ≤ 0.1 were evaluated using the Metascape platform [83]. Cytokines correlated up to *p* ≤ 0.1 were analyzed against each other. The remaining cytokines of the 80 cytokines measured that were quantified, but not correlated to GCS or GOS-E, were used as background subtraction. Protein–protein interactions were considered significant at values of Log10 (*P*) = −2.0 or less by the Metascape platform, with significance increasing as the values decreased.

### 4.11. Ingenuity Pathway Analysis (IPA)

Per Qiagen format, we state that data were analyzed through the use of IPA (QIAGEN Inc., (https://www.qiagenbioinformatics.com/products/ingenuitypathway-analysis, accessed on 14 February 2021)) [84]. We used the cytokine array scores from each patient and healthy donor from plasma, EV lumen, and EV surface cytokines. These were quantitatively analyzed by Core Analyses and Comparison Analyses, where we extracted a list of relevant diseases and biofunctions and resulting networks. For Comparison Analyses, we compared patient vs. healthy donor values (log(2) scale), and the heat map displays activation z-scores for patients relative to controls. For the network analyses, interactome quantitation’s are based on patient vs. healthy donor values (log(2) scale), and network scores are derived from Fisher′s exact test and are −log (*p* values) indicating the probabilities of random protein associations.

### 4.12. Statistical Analysis and Graphics

The statistical analysis was performed using the R suite, version 3.6 [85]. All averages are reported with a standard deviation. Pearson correlations were computed using the Hmisc package in R. Figures were generated using ggplot2 [83]. Univariate comparisons for individual cytokines were assessed using the non-parametric Mann–Whitey-U test. For all tests, the alpha was set to 0.05, and all values between 0.1–0.051 to be trending toward significance. Where applicable, the comparisons were made between patients and control subjects.

## 5. Conclusions

The inflammatory response to stroke involves a series of highly complex interactions of cytokine signaling. Further heightening this complexity is the role of extracellular vesicles in delivery of these cytokines. This investigation sought to provide a comprehensive analysis of cytokine profiles delivered by EVs after hemorrhagic stroke. We identified several EV-contained cytokines that persistently correlate to poor neurological condition throughout stroke recovery. Our results also implicate the cytokine-enriched EV as a highly inflammatory agent detrimental to patient recovery after hemorrhagic stroke. Development of future therapies for hemorrhagic stroke should consider profiling these cell–cell communicators as a comprehensive unit, with EVs as a vehicle in which cytokines may be transported beyond their typical diffusion range. The impact of cytokines assisted by EV transport after CVA could prove to be a critical component in understanding the neuroinflammatory response after hemorrhagic stroke. Continued study of the potential inflammatory impact of EV-trafficked cytokines is necessary to further clarify their role in hemorrhagic stroke recovery.

## Figures and Tables

**Figure 1 ijms-22-07847-f001:**
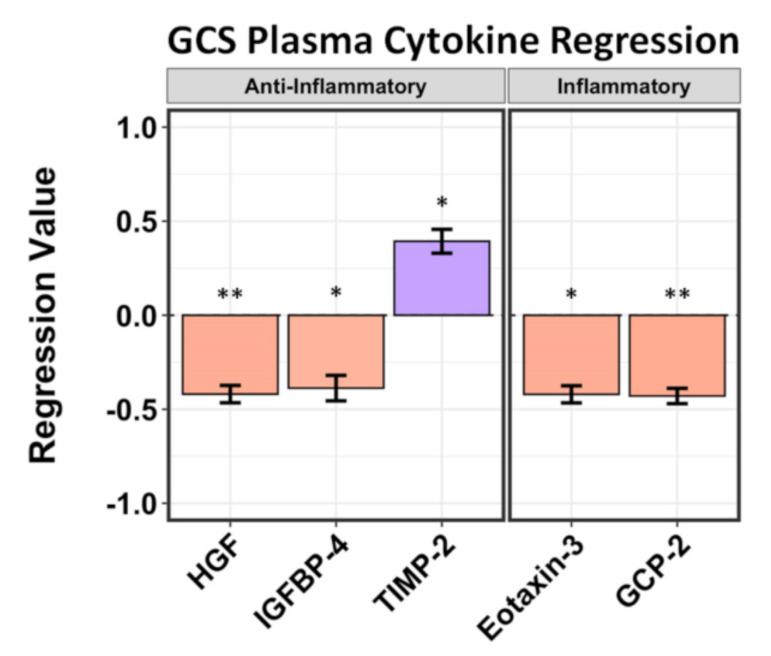
Regression analysis of plasma cytokines correlated to GCS at 5–7 days after hemorrhagic stroke. Negative values correspond to lower GCS; positive values correspond to higher GCS. Three anti-inflammatory cytokines were identified. HGF (*r* = −0.420) and IGFBP-4 (*r* = −0.387) correlate to negative GCS scores, whereas TIMP-2 (*r* = 0.393) correlates to a positive GCS score. The 2 inflammatory cytokines identified, Eotaxin-3/CCL26 (*r* = −0.421) and GCP-2/CXCL6 (*r* = −0.340), both correlate to negative GCS. Asterisks above figure bars indicate statistical significance. One asterisk (*) indicates *p* value smaller than 0.1 (*p* < 0.1). Two asterisks (**) indicate *p* value smaller than 0.05 (*p* < 0.05).

**Figure 2 ijms-22-07847-f002:**
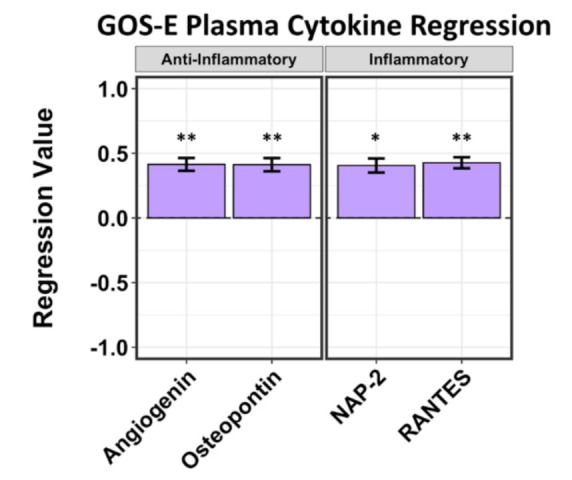
Regression analysis of plasma cytokines correlated to GOS-E at 5–7 days after hemorrhagic stroke. Negative values correspond to lower GOS-E; positive values correspond to higher GOS-E. Four plasma cytokines, including anti-inflammatory angiogenin (*r* = 0.414) and osteopontin/SPP1 (*r* = 0.412) and pro-inflammatory NAP-2/CXCL7 (*r* = 0.405) and RANTES/CCL5 (*r* = 0.426), correlated to positive GOS-E scores. Asterisks above figure bars indicate statistical significance. One asterisk (*) indicates *p* value smaller than 0.1 (*p* < 0.1). Two asterisks (**) indicate *p* value smaller than 0.05 (*p* < 0.05).

**Figure 3 ijms-22-07847-f003:**
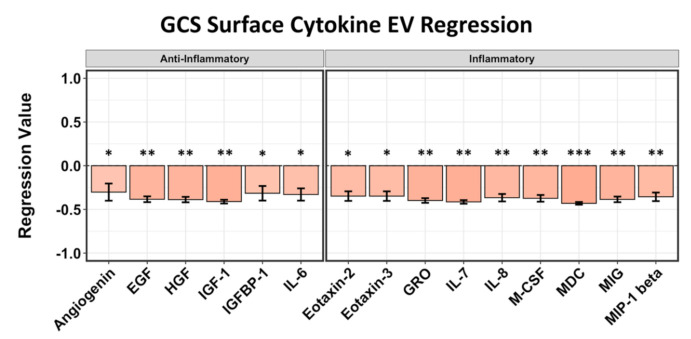
Regression analysis of surface-bound EV cytokines correlated to GCS at 5–7 days after stroke. Six anti-inflammatory and 9 pro-inflammatory cytokines all show negative values corresponding to lower GCS. Asterisks above figure bars indicate statistical significance. One asterisk (*) indicates *p* value smaller than 0.1 (*p* < 0.1). Two asterisks (**) indicate *p* value smaller than 0.05 (*p* < 0.05). Three asterisks (***) indicate *p* value smaller than 0.02 (*p* < 0.02).

**Figure 4 ijms-22-07847-f004:**
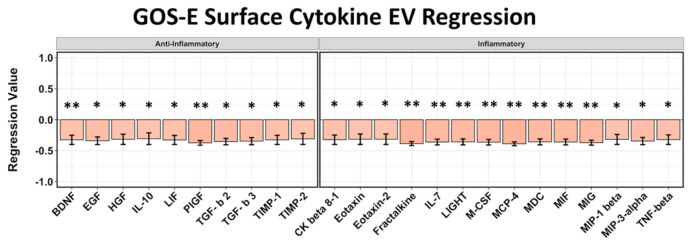
Regression analysis of surface-bound EV cytokines correlated to GOS-E (30+ day outcome) from blood samples taken at 5–7 days after stroke. Ten anti-inflammatory and 14 pro-inflammatory cytokines all show negative values corresponding to lower GOS-E. Asterisks above figure bars indicate statistical significance. One asterisk (*) indicates *p* value smaller than 0.1 (*p* < 0.1). Two asterisks (**) indicates *p* value smaller than 0.05 (*p* < 0.05).

**Figure 5 ijms-22-07847-f005:**
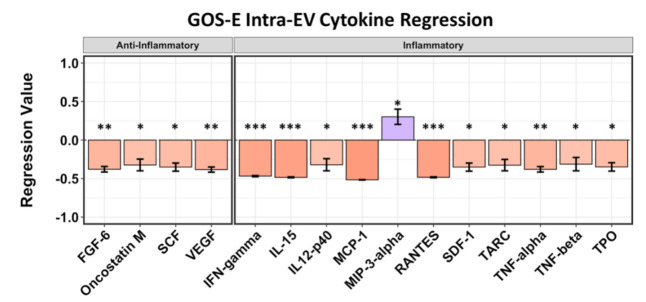
Regression analysis of intra-EV cytokines (after stripping of EV membrane-bound/associated proteins) at 5–7 days after stroke correlated to GOS-E (30+ days after stroke). All 4 of the anti-inflammatory cytokines identified as significant or trending toward significance were correlated to negative GOS-E. Of the 11 pro-inflammatory cytokines identified as significant or trending toward significance, 10 show negative values corresponding to lower GOS-E; cytokine MIP-3 alpha/CCL20 was correlated to a positive GOS-E score (*r* = 0.302). Asterisks above figure bars indicate statistical significance. One asterisk (*) indicates *p* value smaller than 0.1 (*p* < 0.1). Two asterisks (**) indicate *p* value smaller than 0.05 (*p* < 0.05). Three asterisks (***) indicate *p* value smaller than 0.02 (*p* < 0.02).

**Figure 6 ijms-22-07847-f006:**
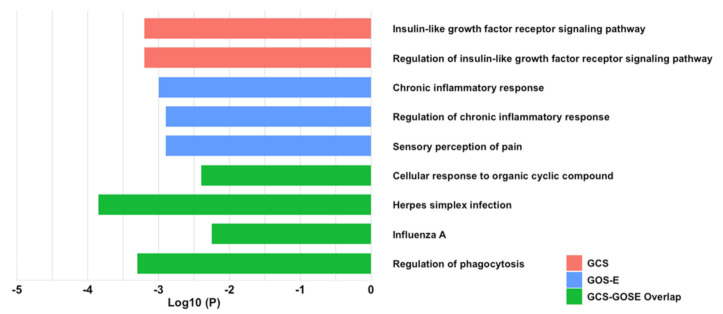
Protein pathway enrichment analysis and protein–protein interactions of cytokines correlated to GCS and GOS-E were analyzed through the Metascape platform. EV cytokines identified as significant or trending toward significance were analyzed, and cytokines not identified were used as background subtraction. Pathway and protein–protein interactions are considered significant at ≤−2.0 Log10 (*P*) by the Metascape platform. Metascape analysis revealed significant interactions with EV surface and intra-EV cytokines correlated to GCS scores at 5–7 days after stroke, including the insulin-like growth factor receptor signaling pathway (Log10 (*P*) = −2.2) and its regulation (Log10 (*P*) = −2.2). EV surface and intra-EV cytokines that were correlated to GOS-E scores revealed interactions relating to the chronic inflammatory response (Log10 (*P*) = −3.0) and regulation of the chronic inflammatory response (Log10 (*P*) = −2.9). Sensory perception of pain was also identified (Log10 (*P*) = −2.9). Cytokines that were only identified as significant or trending toward significance within the plasma were also analyzed but did not yield any additional interactions through Metascape. EV surface and intra-EV cytokines that were correlated to both GCS and GOS-E (MDC/CCL2, M-CSF/CSF1, IL7, and MIG/CXCL9) generated common protein–protein interactions with enrichment clusters for influenza A (Log10 (*P*) = −2.25) and herpes simplex infection (Log10 (*P*) = −3.85). The regulation of phagocytosis (Log10 (*P*) = −3.3) was also correlated. Analysis of all cytokines from both EVs and plasma correlated to GCS and GOS-E did not yield any additional interactions.

**Figure 7 ijms-22-07847-f007:**
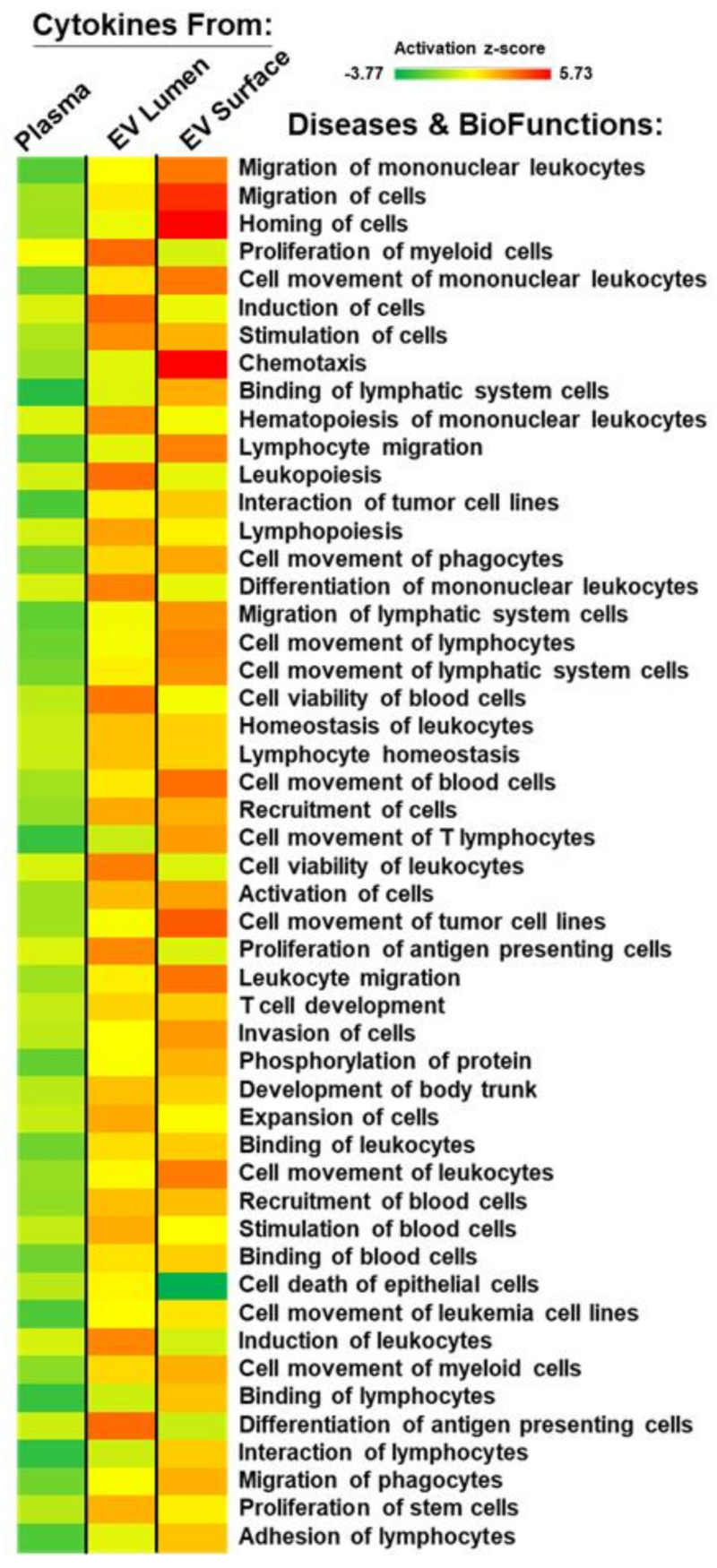
Ingenuity Pathway Analysis/Comparison Analysis shows enrichment of categories related to myeloid cell and lymphocyte movement/migration, homing, proliferation, induction, and activation. Patient and healthy donor cytokine array scores (plasma, EV lumen, EV surface) were analyzed to compare patient vs. healthy donor values (log(2) scale). Heat map was generated from activation z-scores yielding the top 50 diseases and biofunctions.

**Figure 8 ijms-22-07847-f008:**
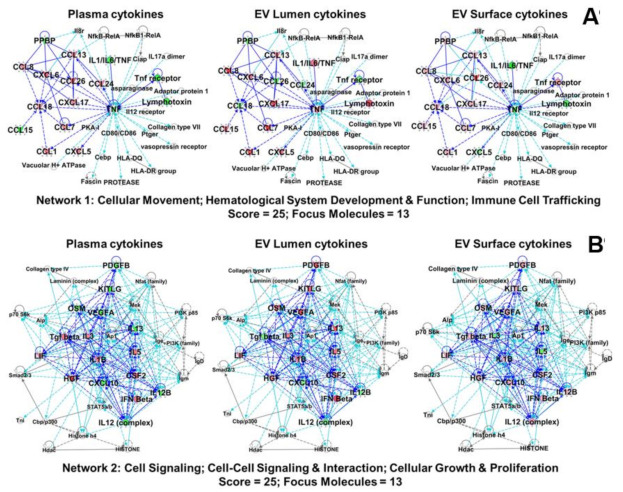
Ingenuity Core Analyses/Comparison Analyses generated networks (top 2 shown). The top 2 derived networks (out of 14; network scores from 25 down to 1) using IPA-extracted pathway data are shown. Network scores are derived from a right-tailed Fisher’s exact test, where scores as *p*-score = −log10 (*p* values) indicate probabilities of random protein associations based on focus molecules in a total gene group. Focus molecules are nodes from which networks initiate. Higher cytokine scores are indicated by red shading and lower values by green shading, relative to their representation in stroke patients vs. healthy donors. Proteins clustered within the top networks/associated functions as derived from IPA algorithms are shown as members of “interactomes”. Proteins from the arrays are labeled in larger bold font, with the background color described above. Solid lines (edges) = direct connections between/among proteins; dashed lines = indirect interactions. Dark blue lines = connections between array proteins; light blue lines = known interactions between array proteins and other proteins in the interactome derived from the Ingenuity Knowledgebase. (**A**) Network 1, “Cellular Movement; Hematological System Development & Function; Immune Cell Trafficking”. (**B**) Network 2, “Cell Signaling; Cell-Cell Signaling & Interaction; Cellular Growth & Proliferation”.

## Data Availability

Data are contained within the article or Appendix A; additional requests for information should be directed to the corresponding authors.

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
