# Peer review of "Cytokine-Laden Extracellular Vesicles Predict Patient Prognosis after Cerebrovascular Accident"

_ijms, 2021, doi:10.3390/ijms22157847_

Round 1

Reviewer 1 Report

Minor Compulsory Revisions:

This is a research paper on the “Cytokine Laden Extracellular Vesicles Predict Patient Prognosis after Cerebrovascular Accident”. The authors are experts in this field and have done a nice job. The paper is clearly and concisely written. However, there are few suggestions that improve it before publication.

  • Author must provide list of Abbreviations that all are used in manuscript for readers. It is difficult to follow short forms in manuscript if full list of Abbreviations is not available.

  • On page 2-3, Author state about that PATIENT numbers in words on page 3, line no 101…

       When on page 2 line no 87 start with 31 patients

       So, be consistent with using numbers for patients total, no words number…

  • On page 13-16, in methods section, Author must be consistent with using numbers or word name of numbers.

When Author mentioned about time, sample quantity then use number not words for that e.g.  On page 14, line no 525 states that for each sample, eight µl was applied to a copper

Above for quantity of sample always use numbers 8 µl not word eight µl.

Same way for minutes on Page 14 line 527, 2 minutes.

  • Check and correct all places of manuscript mainly methods section for above details mainly for patients, sample unit & time and correct it.
  • On page 15, line no 590 under section 4.10 Author stat that outcome to up to p=0.1 were evaluated using the Metascape platform.

          Author must checked for that it is always P< 0.1 not P=.1 for met analysis.

Author Response

Comments and Suggestions for Authors

Minor Compulsory Revisions:

This is a research paper on the “Cytokine Laden Extracellular Vesicles Predict Patient Prognosis after Cerebrovascular Accident”. The authors are experts in this field and have done a nice job. The paper is clearly and concisely written. However, there are few suggestions that improve it before publication.

We thank the reviewer for the kind words, and for the suggestions that have improved the consistency and readability of the manuscript.

Author must provide list of Abbreviations that all are used in manuscript for readers. It is difficult to follow short forms in manuscript if full list of Abbreviations is not available.

This is an excellent suggestion. We have added a list of abbreviations prior to the References and will allow the editorial staff to re-format it as necessary.

On page 2-3, Author state about that PATIENT numbers in words on page 3, line no 101…

       When on page 2 line no 87 start with 31 patients

       So, be consistent with using numbers for patients total, no words number…

On page 13-16, in methods section, Author must be consistent with using numbers or word name of numbers.

When Author mentioned about time, sample quantity then use number not words for that e.g.  On page 14, line no 525 states that for each sample, eight µl was applied to a copper

Above for quantity of sample always use numbers 8 µl not word eight µl.

Same way for minutes on Page 14 line 527, 2 minutes.

Check and correct all places of manuscript mainly methods section for above details mainly for patients, sample unit & time and correct it.

While conventions for spelling out of numbers differ, our lack of consistency was likely annoying.

In all cases we are now consistently using numerals rather than words for numbers (unless a number starts a sentence, where it is convention to spell out the number). Of course, the copy editor will have final say in this, but for now, we are consistent.

On page 15, line no 590 under section 4.10 Author stat that outcome to up to p=0.1 were evaluated using the Metascape platform.

          Author must checked for that it is always P< 0.1 not P=.1 for met analysis.

Thank you for pointing this out, we have corrected this as well.

Reviewer 2 Report

The manuscript “Cytokine Laden Extracellular Vesicles Predict Patient Prognosis After Cerebrovascular Accident” aims at correlating pro-inflammatory cytokines and anti-inflammatory cytokines from plasma extracellular vesicles collected five to seven days after stroke event, with patient’s recovery (measured by various stroke dependent scales on patient’s admission and three (or more) months after stroke.

Overall the manuscript gives new insight for stroke prognosis, however there are some questions I would like to be clarified.

Major revisions

Question 1. Is edema and stroke size correlated with poor outcome during the acute phase of stroke? (Other variables that may contribute for stroke neurological severity).

Question 2. How values of anti-inflammatory cytokines from plasma extracellular vesicles collected five to seven days after stroke event correlate with GOS-E outcome scale, 3 months after stroke?  Where other variables considered for stroke outcome?

Question 3. Are there any specific treatments that patients were taking after stroke, that may also contribute for stroke outcome? My question comes from the possibility that the inflammatory profile of patients may have changed over the period of time included in this study, for instance, if patients were treated with anti-inflammatory drugs.

Question 4. In the discussion, it is not clear to me what it is a worse stroke outcome (quantification/scores) and how to use levels of cytokines to predict clinical outcome (bad or good prognosis), to be used in the clinics.

Line 579

For readers not familiar with the tests and scores, it would be helpful to have some indication regarding scores and stroke neurological severity (for example a score of 0 – normal to 30 – maximum deficit).  It could be included in the Table S1 legend.

Minor revisions

Line 56. “Administration of IL1 receptor antagonists (…)”

Author Response

Comments to the authors

The manuscript “Cytokine Laden Extracellular Vesicles Predict Patient Prognosis After Cerebrovascular Accident” aims at correlating pro-inflammatory cytokines and anti-inflammatory cytokines from plasma extracellular vesicles collected five to seven days after stroke event, with patient’s recovery (measured by various stroke dependent scales on patient’s admission and three (or more) months after stroke.

Overall the manuscript gives new insight for stroke prognosis, however there are some questions I would like to be clarified.

Thank you for your recognition of the novelty of the work, and in particular, we feel your questions have greatly improved how we convey information in the revised manuscript (answers below).

Major revisions

Question 1. Is edema and stroke size correlated with poor outcome during the acute phase of stroke? (Other variables that may contribute for stroke neurological severity).

This is an excellent point that brought up other items worth mentioning.  In the manuscript we say this (Results section 2.1):

The average hemorrhage volume was 32855.49 mm3 +/- 24403.33 SD, which was negatively correlated to the GCS (p=0.012, r =-0.45) and not significantly correlated to the GOS-E (p=0.88, r = -0.03). The type of hemorrhagic stroke was not observed to affect GCS or GOS-E.

Thus, hemorrhage volume was correlated to GCS, where the score was obtained early in the stroke experience, ie, at hospital admission. At determination of GOS-E (minimally 30 days post stroke event), hemorrhage volume was no longer negatively or positively correlated. However, in the Materials and Methods we mistakenly refer to determination of edema volume rather than hemorrhage volume, which was incorrect, and likely contributed to confusion in this area. We have corrected that in the M&M, and have added a paragraph in the Discussion pertaining to the importance of edema and PHE in particular, to ICH etiology. We note that such measurements of edema were beyond the scope of this manuscript (regarding both the temporal layout and more complicated imaging), but would undoubtedly be a valuable area of future research.

Question 2. How values of anti-inflammatory cytokines from plasma extracellular vesicles collected five to seven days after stroke event correlate with GOS-E outcome scale, 3 months after stroke?  Where other variables considered for stroke outcome?

We gather from this that the Reviewer is asking whether a multivariable analysis could have been performed to include the values for anti-inflammatory cytokines from plasma EVs in the model at the same time with other outcome variables (ie, hemorrhage size, age, sex, ethnicity, etc). This also pertains somewhat to Question 3; in the Results, we have added a paragraph noting that for a relatively small N of patients, and a relatively large number of variables (some of which we do not know, such as specifics of medications), we did not believe we could perform robust multivariable analyses in the context of this work. This is a great suggestion for future directions.

Question 3. Are there any specific treatments that patients were taking after stroke, that may also contribute for stroke outcome? My question comes from the possibility that the inflammatory profile of patients may have changed over the period of time included in this study, for instance, if patients were treated with anti-inflammatory drugs.

The Reviewer brings up an excellent point. We note in Results that our lack of knowledge of individual patient treatments is a limitation in the study. As the Reviewer undoubtedly knows, interventions are directed at maintenance of hemostasis, coagulopathy and platelet targets, blood pressure reduction, and possible surgical procedures, among other things. Thus, every patient may have different experiences which could affect the inflammatory milieu reflected in the plasma/EV cytokine content. Again, this is beyond the scope of this manuscript, but is a trove of information for further studies.

Question 4. In the discussion, it is not clear to me what it is a worse stroke outcome (quantification/scores) and how to use levels of cytokines to predict clinical outcome (bad or good prognosis), to be used in the clinics.

We did not attempt a direct quantification of outcome scores in relation to cytokine levels, but rather, we used the full range of GCS and GOS-E values in the correlations to cytokine levels via regression analyses. Those regressions trended positive or negative, but with no definitive cutoff values. Thus, we also did not attempt to link cytokines with particular outcomes, as this would require more clinical (and perhaps subjective) interpretation (and data access) than we feel qualified to perform. Also, our work is far from a defined biomarker evaluation study, although we do hope to make progress in that direction. We note this in the first and last paragraphs of the Discussion, and in portions of the results where we clarified statements regarding regression trends and outcome scores.

Line 579

For readers not familiar with the tests and scores, it would be helpful to have some indication regarding scores and stroke neurological severity (for example a score of 0 – normal to 30 – maximum deficit).  It could be included in the Table S1 legend.

This is another good suggestion from the Reviewer. We have added supplemental Table S2 (with legend and references) to provide context for GCS and GOS-E scores.

Minor revisions

Line 56. “Administration of IL1 receptor antagonists (…)”

Despite many, many eyes having gone over this manuscript, the Reviewer is the first person to catch this. We have changed the wording appropriately.

Reviewer 3 Report

Dear Authors,

I have had the pleasure of reviewing your article: Cytokine Laden Extracellular Vesicles Predict Patient Prognosis 2 After Cerebrovascular Accident

The work is very transparent and interesting. I think this is a topic worth discussing. Well planned work. The summary is clear. Well-chosen title. The article is worth publishing.

Author Response

Dear Authors,

I have had the pleasure of reviewing your article: Cytokine Laden Extracellular Vesicles Predict Patient Prognosis 2 After Cerebrovascular Accident

The work is very transparent and interesting. I think this is a topic worth discussing. Well planned work. The summary is clear. Well-chosen title. The article is worth publishing.

Thank you for your endorsement of the manuscript; we appreciate the time you took to review it, and are delighted that found it worthy of publication.

Round 2

Reviewer 2 Report

The authors of the manuscript “Cytokine Laden Extracellular Vesicles Predict Patient Prognosis After Cerebrovascular Accident” took in consideration all my points and gave very clear answers for my questions.

It is really difficult to use a specific parameter, to correlate with patient's prognosis, since stroke in humans (ischemic and hemorragic) is so heterogeneous and outcome is dependent on other individual comorbidities. I completely agree with authors statement regarding my last question: this would require more clinical (and perhaps subjective) interpretation (and data access).

So, my last suggestion is to change the title. When I first saw the title, as a reader, I was expecting a direct prediction of outcome scores in relation to cytokine levels. In my opinion “Cytokine Laden Extracellular Vesicles Correlates with Patient Prognosis After Cerebrovascular Accident” would be more appropiate title. However, I will leave this suggestion open to the authors, to change it or not.

Lastly, I congratulate the authors for the work presented.